# Intraphagosomal Free Ca^2+^ Changes during Phagocytosis

**DOI:** 10.3390/ijms25084254

**Published:** 2024-04-11

**Authors:** Sharon Dewitt, Joanna Green, Iraj Laffafian, Kimberly J. Lewis, Maurice B. Hallett

**Affiliations:** 1Biomaterials Group, School of Dentistry, College of Biological Life Sciences, Cardiff University, Cardiff CF14 4XY, UK; dewitt@cardiff.ac.uk; 2Neutrophil Signalling Group, School of Medicine, College of Biological Life Sciences, Cardiff University, Cardiff CF14 4XN, UK

**Keywords:** intra-phagosomal Ca^2+^, phagocytosis, Ca^2+^ channels, phagosome, neutrophil

## Abstract

Phagocytosis (and endocytosis) is an unusual cellular process that results in the formation of a novel subcellular organelle, the phagosome. This phagosome contains not only the internalised target of phagocytosis but also the external medium, creating a new border between extracellular and intracellular environments. The boundary at the plasma membrane is, of course, tightly controlled and exploited in ionic cell signalling events. Although there has been much work on the control of phagocytosis by ions, notably, Ca^2+^ ions influxing across the plasma membrane, increasing our understanding of the mechanism enormously, very little work has been done exploring the phagosome/cytosol boundary. In this paper, we explored the changes in the intra-phagosomal Ca^2+^ ion content that occur during phagocytosis and phagosome formation in human neutrophils. Measuring Ca^2+^ ion concentration in the phagosome is potentially prone to artefacts as the intra-phagosomal environment experiences changes in pH and oxidation. However, by excluding such artefacts, we conclude that there are open Ca^2+^ channels on the phagosome that allow Ca^2+^ ions to “drain” into the surrounding cytosol. This conclusion was confirmed by monitoring the translocation of the intracellularly expressed YFP-tagged C2 domain of PKC-γ. This approach marked regions of membrane at which Ca^2+^ influx occurred, the earliest being the phagocytic cup, and then the whole cell. This paper therefore presents data that have novel implications for understanding phagocytic Ca^2+^ signalling events, such as peri-phagosomal Ca^2+^ hotspots, and other phenomena.

## 1. Introduction

Phagocytosis, the process whereby an extracellular object is internalised by a cell (phagocyte), is a complex process involving the extension of small pseudopodia that attach to the extracellular target, initially immobilising the particle within a “phagocytic cup”, and then, after a large cytosolic Ca^2+^ signal, a rapid extension of the pseudopodia fully encloses the target and draws it into the cell interior [1,2,3]. This results in the formation of an intracellular vesicle, the phagosome [4,5], whose membrane originates from the plasma membrane. The relationship between cytosolic Ca^2+^ signalling and phagocytosis is complex, with the cytosolic Ca^2+^ signal that occurs during phagocytosis by neutrophils regulating both the rate of extension of pseudopodia around the target [1,2,3] and the activation of the bactericidal oxidase system in the phagosomal membrane [6,7]. Ca^2+^ signalling within these cells involves both Ca^2+^ influx from the extracellular environment and the release from stored Ca^2+^ from within the cell [8]. Hotspots of Ca^2+^ in the cytosol near the closed phagosome have also been reported in some phagocytic cells such as mouse embryonic fibroblasts and dHL60 cells [9] and RAW 264.7 cells [10]. It is suggested that these Ca^2+^ microdomains (hotspots) arise from “leaky” interactions of STIM-1 on the endoplasmic reticulum (ER), the Ca^2+^ storage organelle with channels on the phagosomal membrane [9]. These STIM-1-mediated Ca^2+^ hotspots are necessary for efficient phagocytosis and oxidase activation [11,12]. In addition, the phagosome itself may contribute to the increase in peri-phagosome Ca^2+^. However, very little is known about the intra-phagosomal Ca^2+^ concentration and how it changes during phagocytosis. Pioneering work by Dahlgren’s group more than 20 years ago [13] using fura2 as a Ca^2+^ indicator showed that intra-phagosomal Ca^2+^ rapidly decreased during phagocytosis. However, the kinetics and relationship to the process of phagocytosis were not investigated. The Ca^2+^ probe used, fura 2, was susceptible to “bleaching effects” [14], making measurements within the phagosome difficult [13]. However, advances in imaging and Ca^2^-sensing fluor design have enabled these early observations to be extended and to further investigate intra-phagosomal Ca^2+^ changes. In this paper, we used a Ca^2+^-sensor, fluo4, which has the more robust fluorescein-like component, covalently coupled to a phagocytic target. When phagocytosed by neutrophils, this indicator dynamically reported the intra-phagosomal Ca^2+^ concentration and enabled a correlate of changes in phagosomal Ca^2+^ with binding, uptake, and phagosomal closure. Using this approach, we report here that there was an initially high Ca^2+^ concentration within the phagosome, which was reduced to the cytosolic concentration within 200 s of phagosomal closure. This is consistent with the incorporation into the phagosomal membrane of opened plasma membrane Ca^2+^ channel, which, in the open phagocytic cup acts as a conduit for local Ca^2+^ elevation, but once the phagosome is closed, “drains” phagosomal Ca^2+^ into the cytosol.

## 2. Results

### 2.1. Fluo4-Zymosan Targets Report Ca^2+^ Concentration

In order to monitor intra-phagosomal Ca^2+^, the Ca^2+^-sensitive fluorescent probe, fluo4, was coupled to C3bi-opsonised zymosan particles as a phagocytic target. The sensitivity of the fluo4-zymosan particles to Ca^2+^ concentration was established in vitro by measuring the fluo4 intensity at various Ca^2+^ concentrations (Figure 1). There were two micro-anatomical zones of the zymosan particle [15], the outer zone and the central core, with the central core having a brighter fluorescence signal as a result of the fluo4 plus a strong auto-fluorescent signal. The fluorescence intensity of the outer zone acted as a true Ca^2+^ indicator, with little autofluorescence. The central fluorescent signal was, however, useful for identifying the location of the target during phagocytosis. Each zone also had a different apparent kd for Ca^2+^ (Figure 1), with the fluo4 in the core zone having an apparent kd of approximately 300 nM, similar to that reported for free fluo4, i.e., 345 nM at pH 7.2, [16], whereas the fluo4 in the outer zone had a higher apparent kd of approx. 1 μM Ca^2+^ (Figure 1). However, in both zones of the zymosan particle, fluo4 reached maximum intensity at 1.3 mM Ca^2+^ (i.e., the extracellular Ca^2+^ concentration) and near-zero intensity at 100 nM (i.e., the cytosol-free Ca^2+^ concentration) (Figure 1). The difference in apparent Kds, fortuitously, also provided the opportunity for a double estimation of intra-phagosomal Ca^2+^ to be carried out from the same zymosan particle within the same phagosome. It should be noted that the fluo4 at both locations was almost totally saturated with Ca^2+^ above 10 μM, so the effective measurement range was thus below 10 μM Ca^2+^.

### 2.2. Phagosomal Fluo4 Signal Is Decreased

Having established that the fluo4-zymosan particles reported Ca^2+^ concentration, the effect of phagocytosis was examined. The fluorescence of zymosan particles within phagosomes was significantly reduced after internalisation compared with non-phagocytosed particles in the same microscopic field (Figure 2a,c). It is unlikely that this was an optical effect caused by the interference from the cytoplasm surrounding the phagocytic target because, (i) during the process of internalisation, no partial or zonal reduction in the fluorescent signal from the zymosan was observed, as would be expected from optical obstruction by the encroaching pseudopodia, and (ii) although there is an effect of light scattering through the cytoplasm that attenuates the detected fluorescent signals from intracellular fluors [17], this would be very small. The light-scattering effect of cytoplasmic projections covering the phagosome depends on the number of cytosolic granules per unit volume (density), the granule refractive index, and their size [18,19]. The engulfing pseudopodia were thin and excluded light-scattering granules, and the cytosolic fluorescence at these locations often has increased excitation efficiency and appears brighter [17]. In a worst-case scenario, taking the published values for bulk cytosolic granules (density, refractive index, and size), Mie scattering calculations provided an estimate of the attenuation coefficient for 488 nm laser light through a neutrophil cytoplasm of about 700 mm^−1^. As the approximate thickness of the cytoplasm covering the zymosan particle within the phagosome was less than 0.5 mm, less than 30% of the laser light entering the pseudopodia would have been scattered and thus would have been unavailable for excitation of fluo4, i.e., the intensity would have been a minimum of 70% of its maximum. This is insufficient to account for the reduction in intensity of the internalised fluo4-zymosan, which was less than about 5% of the non-internalised particles (Figure 2c). Thus, such a decrease in intensity in the phagosomal signal (>95%) cannot be accounted for by an optical effect caused by the attenuation of light-scattering effects of the pseudopodal cytoplasm. On the contrary, the efficiency of excitation and emission of fluors in pseudopodia and near the forming phagosome was increased [17]. The elimination of the possibility that an optical effect decreased the fluo4 signal was confirmed by the use of a non-Ca^2+^ sensitive fluor (fluorescein) conjugated to zymosan, and unlike fluo4-zymosan, the intensity of the fluorescein was not significantly reduced when internalised (Figure 2b,c).

Other possible causes of the reduced emission of phagosomal fluo4 were also excluded (Figure 3b). Changes in the phagosomal pH caused by activation of the oxidase/proton pump system [20] have been shown to cause a small reduction (approx. 10%) in the fluorescence of non-Ca^2+^-sensing but pH-sensitive probe FITC-zymosan at the time when the fluo4-zymosan fluorescence is almost zero. At longer times, when the intra-phagosomal pH falls significantly, a similar reduction in fluorescence of intra-phagosomal fluo4-zymosan has also been observed. This, however, has only been evident at longer times after internalisation (5–10 min), when the intra-phagosomal pH decreased significantly, stabilising at about pH 5 after 15–20 min [19]. Interestingly, the intensity of the fluo4-zymosan core was often seen to increase at low pH (Figure 3b). Thus, these effects of pH cannot account for the decrease in fluo4 fluorescence observed within 0–30 s after internalisation. It was thus concluded that pH changes could not explain the decrease in intra-phagosomal fluo4 signal.

### 2.3. Phagosomal Fluo4 Signal Decrease Not Due to Oxidants

In addition to pH changes, the intra-phagosome environment is exposure to oxidants, notably short-lived superoxide which forms semi-stable hydrogen peroxide. These are generated by activation of the phagosomal oxidase at the phagosomal membrane [21]. The oxidative effect of H_2_O_2_ produced is enhanced by myeloperoxidase, a lysosomal enzyme in azurophilic granules of the neutrophil which are released into the phagosome by fusion with the phagosome. We have previously used zymosan-conjugated dichlorodihydrofluorescein (DCDHF) as an indicator of intraphagosomal oxidants [22], and have shown that the oxidation within the phagosome begins after phagosomal closure and continues with similar kinetics to those of the fluo4 intensity decreased reported here. However, no immediate in vitro effect of H_2_O_2_ and myeloperoxidase was observed and fluo4 zymosan intensity was not significantly reduced before approx. 100 s of contact (Figure 3a). Even by 100 s, the intensity was reduced only be approx. 15%. It was thus concluded that intra-phagosomal oxidation could not account for the decrease in fluo4-zymosan intensity observed during phagocytosis. It should also be noted that fluorescein (a chemically related molecule) was also resistant to the bleaching effect of the intra-phagosomal environment as it remained at the same brightness whether outside or inside the phagosome (Figure 2b).

### 2.4. Phagosomal Fluo4 Zymosan Remained Responsive to Ca^2+^ Changes

In order to test whether fluo4 attached to zymosan remained responsive to Ca^2+^ after contact with the intra-phagosomal environment, three approaches were adopted. The first approach was simply to liberate the zymosan particle from within the phagosome by lysis of the cells with Triton X-100 in order to allow contact of the released particles with the extracellular high Ca^2+^ (1.3 mM). This procedure significantly increased the fluorescent signal, consistent with retained Ca^2+^ sensitivity (Figure 3c). As Triton X-100 destroyed the cell morphology, this approach could not be undertaken during single cell imaging. Instead, cell populations were mixed with zymosan particles and centrifuged to bring the particles and cells in close contact, and then, after 5 min at 37 °C, they were resuspended and three cycles of sedimentation at 1 g were used to separate free zymosan particles from cell-associated zymosan. The resultant cell pellet was sampled for imaging to confirm that all zymosan particles were within phagosomes and their fluorescence intensities were measured. Triton X-100 was then added to a portion of the cell pellet and sampled for imaging to confirm that no zymosan particles were within phagosomes and their fluorescence intensity was measured. There was a highly significant increase in the fluorescence intensity of the zymosan after Triton X-100 treatment (Figure 3c). These data are consistent with retention of Ca^2+^ sensitivity by the phagosomal fluo4-zymosan particles, and when released into the high-Ca^2+^ extracellular environment, their fluorescence increased. The second approach was to use saponin (digitonin), a plant detergent-like molecule that selectively permeabilizes the cell membranes [23] and collapses the transmembrane Ca^2+^ gradient, to saturate the intracellular fluorescent Ca^2+^ indicators with Ca^2+^ [24] but leave the cell morphology recognisable [23]. Saponin permeabilisation significantly increased the fluorescent signal from fluo4-zymosan within phagosomes (Figure 3c), confirming that after phagocytosis, fluo4-zymosan retained its Ca^2+^ sensitivity. However, in both the previous approaches, the intra-phagosomal fluo4 zymosan particles were exposed to high concentrations of Ca^2+^. A third approach was therefore also adopted to exclude the possibility of fluo4 remaining only weakly sensitive to Ca^2+^ and responding to only high Ca^2+^. The third approach was to artificially elevate cytosolic Ca^2+^ using a Ca^2+^ ionophore, ionomycin. Elevation of cytosolic Ca^2+^ by ionomycin caused a significant increase in fluo-4 fluorescence from particles within the phagosomes (Figure 3c,d). This was consistent with the increase in intra-phagosomal Ca^2+^ detected by the functional Ca^2+^ probe whilst within the phagosome. The kinetics of the intra-phagosomal Ca^2+^ increase also suggest that the Ca^2+^ channels on the phagosomal membrane were open. However, it cannot be excluded that ionomycin partitioned across all intracellular membranes and that Ca^2+^ entered the phagosome via this route. Whatever the mechanism for the increase in intra-phagosomal Ca^2+^, the effect show that the intra-phagosomal fluo4 remained sensitive to physiological changes in Ca^2+^ concentration. It was thus concluded that the decrease in fluo4 intensity during phagocytosis was the result of a decrease in intra-phagosomal Ca^2+^ concentration from an initially saturating concentration of Ca^2+^ (probably the extracellular concentration of 1.3 mM) to the cytosolic Ca^2+^ concentration (approx. 100 nM).

### 2.5. Kinetics of Intra-Phagosomal Ca^2+^ Decrease

By monitoring the intensity of fluo4-zymosan particles during internalisation by the phagocyte, the relationship of the decrease in intra-phagosomal Ca^2+^ with the stages of phagocytosis were established. On engaging the zymosan particle, the cells form a “phagocytic cup” at the base of the zymosan, and the “cup” then extends until the particle is enclosed within the phagosome [1,25]. Although the time of contact of the particle with the cell could be determined, the exact time of complete phagosome closure was more difficult to establish but could be estimated within a few frames (about 0.2–0.5 s) from phase contrast imaging. At the time of phagosome closure, the intensity of fluo4-zymosan was unchanged and equal to free zymosan particles in high Ca^2+^ (1.3 mM). However, within 5–10 s of closure (6.3 ± 1.7 s: mean ± sd; *n* = 14), the intensity of the internalised fluo4-zymosan particles abruptly decreased, reaching its low equilibrium level within 20 s (Figure 4 and Appendix A). Presumably, the high (millimolar) phagosomal Ca^2+^ at the time of phagosome closure leaked out of the phagosome into the cytosol during this time. Based on knowledge of the phagosome water volume, it can be estimated that the phagosome initially contained about 2 × 10^7^ Ca^2+^ ions, and thus that it was lost at an average rate of about 10^6^ ions/s over 20 s to reduce this to the cytosolic concentration of c.100 nM. From the time when phagosomal Ca^2+^ was reduced to below the fluo4 saturation level (10 μM), there was an approximately linear decrease in fluo4 intensity equivalent to a decrease in intra-phagosomal Ca^2+^ of from 10 μM to 0.1 μM over 5 s, with an approximate half time (t_1/2_) of 2.6 s ± 0.3 s S (mean ± sd; *n* = 14). This was slower that the initial rate of loss expected if the outward leak had been driven by the Ca^2+^ concentration gradient across the phagosomal membrane. In this phase, Ca^2+^ ions would have crossed the phagosomal membrane at a rate of about 0.2 × 10^4^ Ca^2+^ ions/s. Although which Ca^2+^ leak pathways were responsible for the decrease in intraphagosomal Ca^2+^ was not established here, TRPM2 Ca^2+^ channels are found on phagosomes [26] and carry currents of about 1 pA in neutrophils [27] or about 10^7^ ions/s, suggesting that only a few of these channels need to be open on the phagosome membrane to lower the intraphagosomal Ca^2+^ concentration at the rates observed (e.g., 5 channels each open for 0.2 s would drain the phagosomal Ca^2+^ from 1.3 mM to 10 μM in 2 s). It is thus possible that that plasma membrane Ca^2+^ channels opened by the phagocytic stimulus, which gives rise to the well-documented large phagocytic cytosolic Ca^2+^ signal [1,2,22,25], remain open in the phagosomal membrane and thus “drain” the enclosed phagosome of its Ca^2+^.

### 2.6. Subcellular Location of Ca^2+^ Leakage Pathways

In order to test the conclusion that a Ca^2+^ leak occurred across the phagosomal membrane, the translocation of the YFP-tagged C2 domain of PKC-γ (YFP–C2-γ) from the cytosol to the plasma membrane was used as a real-time spatial marker of Ca^2+^ influx [28,29]. However, as human neutrophils cannot be transfected, a phagocytic cell line, RAW 264.7 macrophages, were used. Although it is not entirely satisfactory to extrapolate results from a mouse macrophage cell line to human neutrophils, RAW cells are phagocytic and also show the same decrease in intraphagosomal Ca^2+^ as demonstrated in neutrophils. These cells thus provided a useful model on which to test the previous conclusion. In this cell line, the translocation of YFP–C2-γ to the plasma membrane has been shown to be driven by an influx of Ca^2+^ ions from the extracellular medium [30]. This was achieved by opening store-operated Ca^2+^ channels (using thapsigargin) and then correlating YFP–C2-γ translocation with pulses of extracellular Ca^2+^ applied experimentally [31]. When presented with mouse C3bi-opsonised-zymosan (without fluorescence tags), translocation of YFP–C2-γ to the plasma membrane was observed after phagocytic cup formation, reflecting the large cytosolic Ca^2+^ signal expected to accompany phagocytosis. There was also an initial localised translocation of YFP–C2-γ to the phagocytic cup, indicating that Ca^2+^ leakage pathways were open in this membrane (Figure 5a—1 s). Within the next second (labelled “2 s” in Figure 5a), translocation of YFP–C2-γ was observed at all plasma membrane locations (Figure 5a—2 s). Translocation of YFP–C2-γ to the plasma membrane was observed in all cells undergoing phagocytosis, and localised translocation to the phagocytic cup was observed before the global translocation in 4 out of 15 cells. The rapid opening of Ca^2+^ channels remote from the phagocytic site explains the difficulty in observing localised cytosolic Ca^2+^ increases during phagocytosis, as the global open Ca^2+^ channels overwhelmed the effect of a localised influx of Ca^2+^ [22]. Clearly, Ca^2+^ leakage occurs on the forming phagosome after phagosome closure. In the example shown, the cell also contained a previously formed phagosome (Figure 5b, marked “x”) that initially had no YFP–C2-γ on its surface, indicating the absence of a Ca^2+^ flux from the phagosome. However, when the cytosolic Ca^2+^ signal was triggered by the second phagocytotic event, it was seen that YFP–C2-γ also translocated to the membrane of the previous phagosome (especially obvious at 20 s, in Figure 5c). This can be explained by either (i) the Ca^2+^ leakage pathways on the previously formed phagosome remaining open, so that the elevated cytosolic Ca^2+^ began to refill the previously drained phagosome that began leaking out (detected by YFP–C2-γ translocation), or (ii) closed physiological Ca^2+^ channels being re-opened as result of diffusible Ca^2+^ channel-opening stimuli (e.g., IP_3_). Thus, the use of YFP–C2-γ demonstrates that Ca^2+^ leakage pathways from the forming phagosome were open during phagosome formation and remained open (or, for physiological Ca^2+^ channels, remained operational) after phagosome closure.

## 3. Discussion

The results presented in this paper show that an open phagosome entraps extracellular Ca^2+^ that “empties” into the cytosol once the phagosome has closed. The concentration of phagosomal Ca^2+^ within the phagocyte equilibrates with the cytosolic Ca^2+^ within 20 s of phagosome closure. This sequence is shown as a cartoon (Figure 6). Thus, the phagosome cannot act as a long-term store of Ca^2+^ for later signalling events. Also, it cannot act as a reservoir for loading lysosomes with Ca^2+^ for other signalling events [30]. The rapid decrease in intra-phagosomal Ca^2+^ was reported over 20 years ago by Dahlgren’s group, who monitored Ca^2+^ in a similar way to that described here but using fura2 as the Ca^2+^ reporter. Although fura2 was bleached during the experiments (both 340 and 380 nm decreasing), the group found that the intra-phagosomal Ca^2+^ level dropped rapidly. Little notice has been given to this early observation, which is confirmed (and extended) here. However, Anke et al. [26] more recently showed that in phagosomes from phagocytes taken from *Trpm2^−/−^* mice, i.e., lacking the TRPM2 Ca^2+^ channel, intra-phagosomal Ca^2+^ increased slowly (over a 10 min time period) compared to the intra-phagosomal Ca^2+^ in phagosomes expressing TRPM2. They suggested that the phagosomes devoid of TRPM2 lacked a leakage channel and thus Ca^2+^ accumulated within the phagosome, presumably as a result of plasma membrane Ca^2+^ pumps. The conclusion that open Ca^2+^ channels on the phagosomal membrane are responsible for the loss of intraphagosomal Ca^2+^ cellular is similar to that which was drawn from the data shown here.

From the work presented here, it seems unlikely that Ca^2+^ events could be driven by phagosomal Ca^2+^ close to the time of completion of phagocytosis. However, if the Ca^2+^ channels closed slowly (over minutes rather than seconds) and the phagosomal membrane contained functional plasma membrane Ca^2+^ pumps, it is expected that the phagosome would re-load with Ca^2+^ over a longer time period. The equilibrium generated across the plasma membrane would thus also be generated across the phagosomal membrane. Under these circumstances, phagosomal Ca^2+^ could be important for driving peri-phagosomal Ca^2+^ events. The earliest report of such a Ca^2+^ cloud localised to near the phagosome [32] is usually dismissed as almost certainly an optical artefact caused by the thinner cytoplasm around the pseudopodia that increased the excitation and efficiency in that area [17]. However, the phagosome itself may be a source of Ca^2+^ that adds to or generates peri-phagosomal Ca^2+^ events such as Ca^2+^ hotspots [9,10].

## 4. Materials and Methods

### 4.1. Cell Preparation

Human neutrophils, isolated from the blood of healthy volunteers who had given informed consent, as described previously [10], were suspended in Krebs medium (NaCl 120 mM, KCl, 4.9 mM KH_2_PO_4_, 1.2 mM MgSO_4_, 1.2 mM CaCl_2_, 1.3 mM, HEPES 25 mM, and bovine serum albumin 0.1% adjusted to pH 7.4 with NaOH).

### 4.2. Properties of the Phagocytic Target

Zymosan particles (killed yeast cells) of approximately ellipsoid shape, having radii in the *x*, *y*, and *z* planes (or semi-axes) of 1:1.5 and 1 μm, were opsonised with iC3b by incubation with human or mouse serum as previously described [11]. The particles have a cell wall enclosing an outer space and a dense central core [12]. The water space in the zymosan-containing phagosome is approximately 50% of the total volume [12], and thus each would contain 6.3 μm^3^ of extracellular medium (1.3 mM Ca^2+^). A cadaverine derivative of the fluorescent Ca^2+^ indicator fluo4 (fluo4 cadaverine (F36201 fluo-4 cadaverine, pentapotassium salt, Molecular Probes; Thermo Scientific, Waltham, MA, USA) was linked to amines in the zymosan particles by carbodiimide linkage using (1-ethyl-3-[3-dimethylaminopropyl]carbodiimide hydrochloride, Thermo Scientific). The reagents were used at 1 mg/mL and incubated with a suspension of zymosan particles at room temperature for 4 h before separation of the reactants from the solid particles by 3 rounds of centrifugation and resuspension in fresh medium to remove the reactants. A similar procedure was used to produce non-Ca^2+^-sensing fluorescein-linked zymosan particles by reacting the particles with fluorescein isothiocyanate (FITC, Sigma, Livonia, MI, USA) as described previously [33].

### 4.3. Imaging and Intra-Phagosomal Ca^2+^ Monitoring

The fluo4-zymosan particles were sedimented onto a glass coverslip mounted onto a thermostatically controlled stage (37 ± 0.1 °C) in Krebs medium. The coverslip was then washed with medium to remove particles that had not made contact with the coverslip and were loosely adhered. Neutrophils were then allowed to adhere to the same glass coverslip during microscopic observation, and a microscopic field was selected in which neutrophils and particles were close and expected to undergo phagocytosis. The field was then imaged using a resonant laser scanning head of the LeicaSP5 confocal inverted microscope (Leica Microsystems, Heidelberg, Germany) and with a 63× objective. Fluorescence (excited by 488 nm laser scanning) and transmitted light (phase contrast images) were acquired simultaneously. Image analysis and presentation were achieved using Image J (v.1.38e) software (https://imagej.nih.gov/ij/, accessed on 2 February 2020).

### 4.4. Estimation of Intraphagosomal Ca^2+^ Concentration

The concentration of intra-phagosomal Ca^2+^ was estimated from the fluorescence intensity of the zymosan particles during the experiment (F). The intensity of zymosan particles outside the cell (Zo) in 1.3 mM Ca^2+^ was taken as the Ca^2+^ saturation maximum (Fmax), and the minimum fluorescence (Fmin) was estimated as being the autofluorescence signal from unlabelled zymosan particles, as in the in vitro calibration curves (see Figure 1). Using the kd values from the calibration curves, the intra-phagosomal Ca^2+^ concentration was estimated using a variant of the standard equation for single-wavelength fluors [24]: Ca^2+^ = Kd (α − 1)/(β − α), where α = F/Fmin and β = Fmax/Fmin.

### 4.5. Raw 264.7 Cell Transfection

RAW 264.7 cells were electroporated to introduce the C2-γ-YFP plasmid (3 µg plasmid DNA per 2 × 10^6^ cells) using the Cell Line Nucleofector (Lonza) as described previously [31]. Cells were incubated at 37 °C in 5% CO_2_ for 3–4 h to enable expression of the newly introduced DNA before imaging on a Leica SP5 confocal microscope. Fluorescent protein expression in transfected RAW 264.7 cells was detected by ~1 h post transfection, but expression was optimal at ~4 h post transfection. The plasmid encoding for C2-γ-YFP was a kind gift from Theodorus W. Gadella (Swammerdam Institute for Life Sciences, University of Amsterdam, the Netherlands).

## Figures and Tables

**Figure 1 ijms-25-04254-f001:**
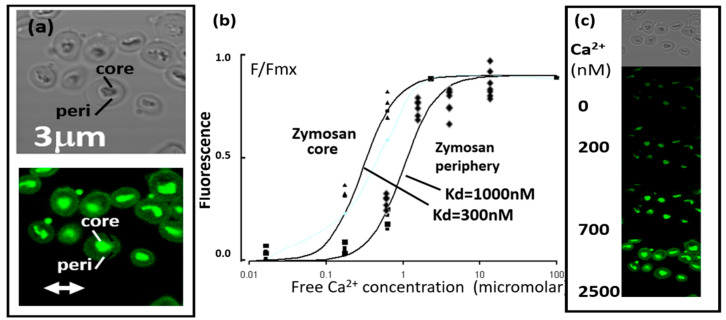
Properties of fluo4-zymosan particles. (**a**) The phase contrast (upper) and fluorescent (lower) appearance of zymosan particles in 1.3 mM Ca^2+^, with the dense “core” and the transparent periphery indicated. (**b**) The relationship between free Ca^2+^ concentration and fluorescence intensity from the core and peripheral regions of individual zymosan particles. Individual experimental data are shown as F/Fmax (with separate symbols), together with the theoretical relationship (lines) shown for two dissociation constants (kd). (**c**) A sample experiment showing the same microscopic field; top = phase contrast; then florescence images in the same field at the Ca^2+^ concentrations indicated.

**Figure 2 ijms-25-04254-f002:**
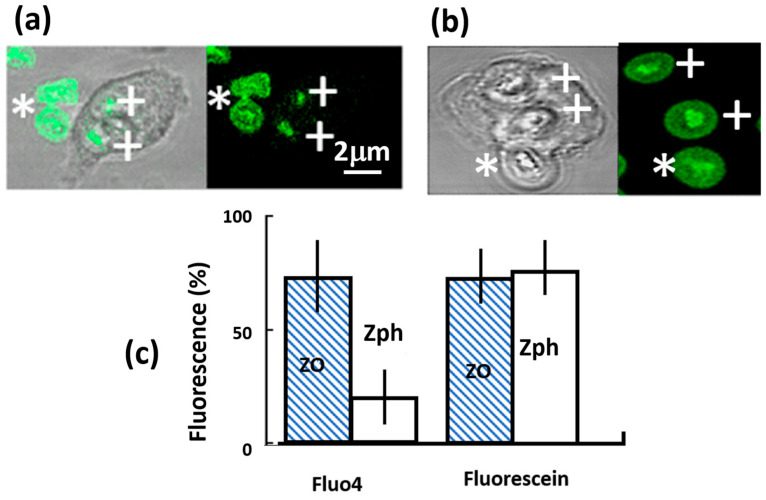
Comparison of fluorescein and fluo4-coupled zymosan particles. In (**a**,**b**), the phase contrast and corresponding fluorescent images are shown for (**a**) fluo4-coupled zymosan and (**b**) fluorescein-coupled zymosan. Both sets of images show the internalised particles, marked by a “+” and external particles by a “*”. (**c**) The quantitation of fluorescence from fluo4 and fluorescein particles either outside the cell (ZO) or inside a phagosome (Zph). The fluorescence units were arbitrary but were comparable, as measurements were taken from the same microscopic fields with the same excitation strength and detection sensitivity. The bars show the mean and the vertical line shows the range for at least 50 determinations.

**Figure 3 ijms-25-04254-f003:**
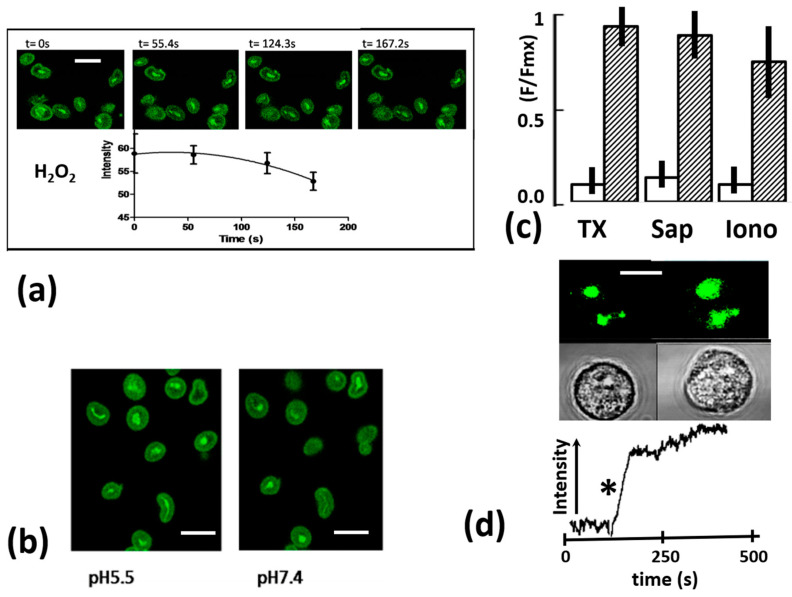
The effect of experimental manipulations on fluo4-zymosan fluorescence. (**a**) The effect of H_2_O_2_ on fluo4-zymosan fluorescence incubated for the lengths of time shown. The top panel shows a typical experiment and the graph below shows the combined data from 3 separate experiments (*n* = 100 particles). The bars show the S.E.M. for the 3 experiments. (**b**) A typical experiment demonstrating the effect of acidification on fluo4-zymosan-fluorescence at the pH shown. In repeat experiments, fluo4 intensity in the zymosan periphery was reduced by 8.5 ± 0.7% and the core increased by 2.2 ± 1.9% when the pH was reduced from 7.5 to 5.5 (mean ± sem., *n* = 25). (**c**) The intensity of fluo4-zymosan within the phagosome (open bars) and after treatment (cross-hatched bars). The pair marked “TX” is before and after TritonX-100 (0.1%) treatment; the pair marked “Sap” is before and after saponin (10% *w*/*v*) treatment; the pair marked “Iono” is for the same zymosan particles before and after the addition of ionomycin (1 μM). In each case, the bars show the mean and the vertical lines the range of replicate experiments for 5 experiments. (**d**) The time course of the change in fluo4-zymosan intensity after the addition of ionomycin, with the asterisk indicating the time point at which ionomycin was added and the images above the same cell depicting two fluo4-zymosan particles before and after ionomycin treatment. The scales bars on each image represents 2 μm.

**Figure 4 ijms-25-04254-f004:**
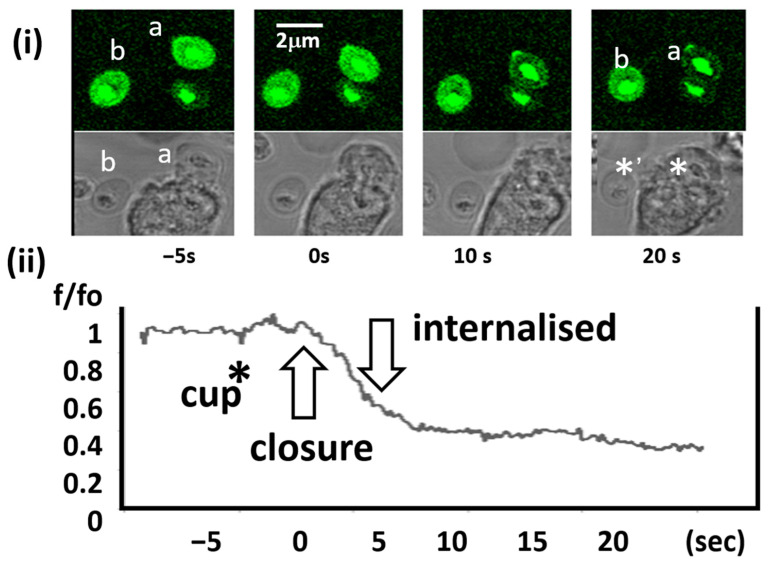
Changes in fluo4-zymosan intensity during phagocytosis. (**i**) Fluorescence and phase contrast images of the time sequence of a neutrophil as it internalises a zymosan particle. The particle that is internalised is labelled (a) and (b) is the “control” particle, which remains external. Each image shows the progress of phagocytosis at the times indicated. The location of particles a.b are indicated on the phase contrast image as * and *’ respectively. (**ii**) The lower graph shows the complete time sequence of the decrease in fluo4-zymosan intensity with key events (cup formation, closure of the phagosome, and internalisation) marked. The sequence is typical of at least 14 other phagocytotic events. The movie in the Appendix A shows a different experiment, in which a similar complete process of phagocytosis and the accompanying decrease in fluo4-zymosan fluorescence can be seen.

**Figure 5 ijms-25-04254-f005:**
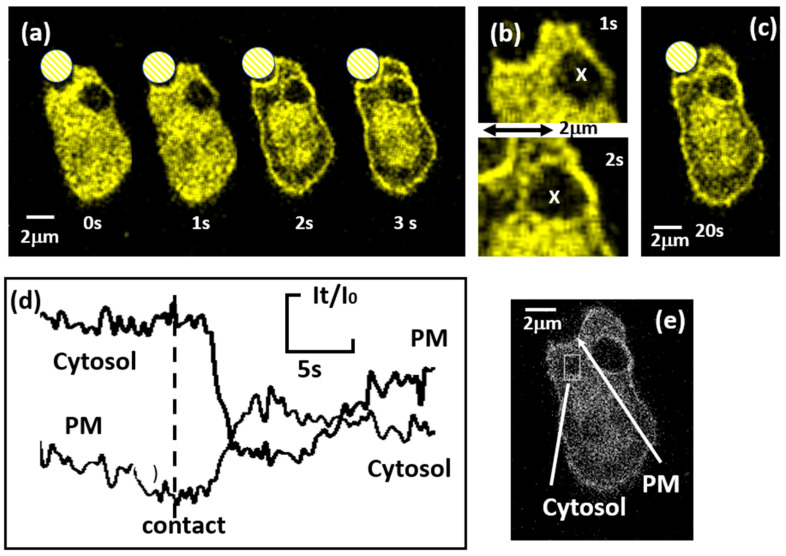
Sites of Ca^2+^ influx marked by YFP-tagged C2 domain of PKC-γ. (**a**) A sequence of images of a RAW 264.7 cell expressing YFP–C2-γ. The phagocytic target is indicated by the crosshatched circle. At time zero (as indicated), a phagocytic cup formed at the base of the zymosan without translocation of YFP–C2-γ from the cytosol. After 1 s, translocation of YFP–C2-γ from the cytosol to the plasma membrane of the phagocytic cup is seen. This is also shown in the enlarged image (**b**). At 2 s, translocation of YFP–C2-γ to the phagocytic cup and the rest of the plasma membrane is obvious, as can also be seen in the image marked “3 s”. (**b**) Enlarged images showing a previous old phagosome marked by an “X”, which has no translocated YFP–C2-γ on its membrane at 1 s, but in the lower image at 2 s, some translation had occurred. In image (**c**) at 20 s, translocation of YFP–C2-γ had remained globally on the plasma membrane and can also be seen more clearly on the previously internalised phagosome. (**d**) The time course of the relative intensity changes measured at time t (It) as a fraction of the intensity at time zero (I_0_), measured within the measurements zones indicated at the plasma membrane (PM) and cytosol (within the box) indicated in (**e**).

**Figure 6 ijms-25-04254-f006:**
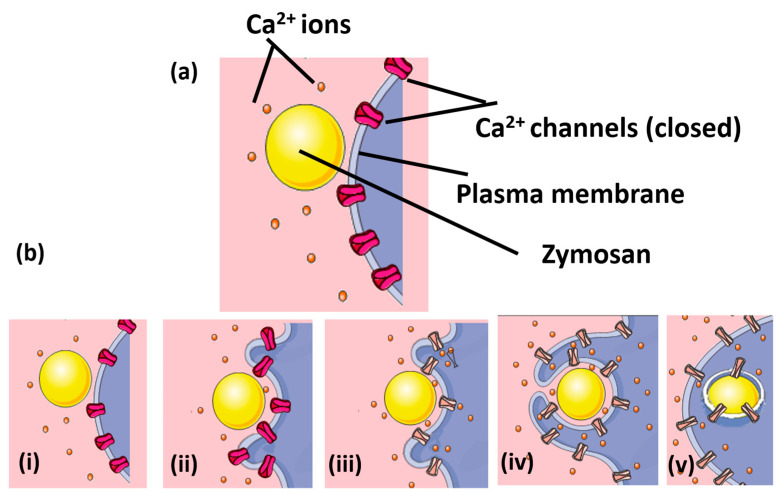
Proposed sequence of Ca^2+^ channel opening on the phagosomal membrane. (**a**) A “cartoon” of some of the components of the proposed model system. The phagocyte plasma membrane with Ca^2+^ ion channels (closed) before contact with the phagocytic target (zymosan particle) and Ca^2+^ ions is labelled. (**b**) The proposed sequence of events during phagocytosis are shown as follows. (**i**) Before contact between the phagocyte and the target, where the cytosolic free Ca^2+^ ion concentration is low. (**ii**) Contact between the particle and the phagocyte, resulting in the formation of a phagocytic cup, with the Ca^2+^ ion channels remaining closed and the cytosolic free Ca^2+^ ion concentration still low. (**iii**) A critical number of receptors are engaged, the Ca^2+^ channels remain open, and the cytosolic free Ca^2+^ ion concentration begins to rise. Note that the location of the open Ca^2+^ channels includes the portion of plasma membrane that forms the base of the forming phagosome and thus provides a mechanism for preferentially elevating Ca^2+^ near the forming phagosome. (**iv**) In response to the elevated cytosolic free Ca^2+^ ion concentration, the pseudopodia around the target nearly enclose it, trapping Ca^2+^ ions in the forming phagosome. (**v**) Phagocytosis is complete but Ca^2+^ channels remain open on the phagosomal membrane, such that intra-phagosomal Ca^2+^ leaks out into the surrounding cytosol. Once “drained” of Ca^2+^, the phagosome plays no further role in Ca^2+^ signalling.

## Data Availability

Data are contained within the article and Appendix A.

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
