# Peer review of "Intraphagosomal Free Ca2+ Changes during Phagocytosis"

_ijms, 2024, doi:10.3390/ijms25084254_

Round 1

Reviewer 1 Report

Comments and Suggestions for Authors

This study reports measurement of the free calcium concentration within phagocytic vacuoles of human neutrophils and cultured macrophages. Using yeast particles labelled with the calcium-sensitive dye Fluo4, the authors show that the calcium concentration inside vacuoles decreases within 20 seconds following particle internalisation. The fluorescence changes were not due to changes in phagosomal pH or redox state. They further show that a calcium-sensing lipid-binding protein domain is recruited to a macrophage vacuole, indicating that sites of high calcium concentrations form around phagosomes. They conclude that calcium ions are rapidly drained from phagosomes by calcium channels and that this flux contributes to the calcium signals that drive phagocytosis.

Comments:

The study is interesting and the data generally consistent with the conclusions drawn, but the dataset is rather limited, gathered in different cellular systems, and in part overinterpreted. Specifically:

1.       I find it difficult to associate data from primary human neutrophils (Fig 1-4) to data gathered in a mouse macrophage cell line (Fig 5). Human neutrophils and mouse macrophages have a different display of ion channels and engage different signalling pathways during phagocytosis. I understand that primary neutrophils are difficult to transfect, but why were macrophages chosen for the experiments in Fig. 5 and not the HL-60 cell line that can be differentiated into the neutrophilic lineage? At the minimum, the authors should repeat the Fluo4-zymosan experiments in RAW macrophages to show that they behave as neutrophils.  

 2.       The time-resolved recordings in Figs 3D, 4, and 5 are illustrative examples and lack statistical proof (n=1). Additional recordings are needed to properly document the kinetics of the changes in phagosomal calcium (Fig. 4) and the presence of sites of high Ca2+ concentrations around phagosomes (Fig. 5). 

3.       The images in Fig. 5 do not prove the presence of open calcium channels on phagosomes. The calcium-sensing module used binds to phospholipids in a calcium dependent manner, not to activated calcium channels. The observation that it decorates the membrane of a pre-formed phagosome during the ingestion of another particle does not prove that calcium channels are open on this phagosome. As the authors acknowledge, the entire plasma membrane is decorated by the calcium-sensing module in the image shown. In their 2002 Science paper, Teruel and Meyer report that the sensor is targeted to the plasma membrane transiently when calcium is released from internal stores, and persistently in response to entry of extracellular calcium across the plasma membrane. Fig 5 shows a transient recruitment, which suggests that the calcium released from stores during the engagement of phagocytic receptors causes the transient accumulation of the probe around the plasma and phagosomal membranes. This single image therefore cannot be used as proof to document the presence and activity of calcium channels on the membrane of phagosomes.  The authors should revise their conclusions or provide functional evidence that calcium channels contribute to the drop in phagosomal Ca2+, for instance with inhibitors of TRPM2 (JNJ-28583113) or store-operated channels (GSK-7975A, SKF-96365 or lanthanum chloride).

4.       The bold statement on line 47 that neutrophils have no endoplasmic reticulum is misleading. Compared to dendritic cells indeed neutrophils have much less ER, but they are hardly devoid of it. EM micrographs show plenty of ER membranes. I could not access the reference 13, a book from 1973, but one reference from a book from 1975 is not sufficient foundation for such a bold statement. Many functional studies report a functional ER in neutrophils (studies on ER stress effect in neutrophils, studies on store-operated calcium entry and STIM1/STIM2 function in neutrophils... etc). In Nunes et al 2012 there is electron microscopy evidence of endoplasmic reticulum contacts with phagosomes, documented in mouse wild-type and reduced in STIM1 knock-out primary neutrophils by EM (Figure 2b) as well as immunofluorescene of STIM1 recruitment to periphagosomal sites that is increased by thapsigargin (Fig 1c). This sentence should be revised, the citation of Ref 9 corrected and the summary figure modified to include the ER.

Comments on the Quality of English Language

There are many space typos and grammar errors throughout, could be a formatting issue but would be nice to fix before publication.

Author Response

See our replies to comment on the attached document comment

Reviewer 2 Report

Comments and Suggestions for Authors

Phagocytosis generates phagosomes in which the [Ca2+] may be as high as the extracellular [Ca2+] initially and then declined to the cytosolic [Ca2+]. The authors tested this hypothesis by measuring [Ca2+] in phagosomes during different stages of phagocytosis using an improved fluorescent Ca2+ sensor. After they ruled out several potential trivial causes that may reduce the fluorescence in the phagosomes, they concluded that the reduction of [Ca2+] in the phagosomes is due to Ca2+ channels that are open in the plasma membrane that forms the phagosomal cup, and that remain open after phagosome closure and internalization until the phagosomal [Ca2+] is equal to the cytosolic [Ca2+]. Overall, the presentation of the work is fine, most of the data are clearly shown and rightly interpreted, and the conclusions are mostly evidence based or reasonably argued. The working model presented at the end is interesting and communicates the take home message.

There are some minor problems that should be fixed before publication, as described below.

The Kd's motioned in line 77-80 do not match the data in Fig 1b. What is the [Ca2+] for the images shown in Fig 1a that also lacks a scale bar? There is no Y-axis in Fig 1b. Fig 1c also lacks a scale bar.

The statement in line 117-119 is hard to evaluate because the data in Fig 2c was not presented more quantitatively. The fluorescence intensity in Y-axis were labeled with 0 and 100 without the ticks on the axis. Minor ticks for the 10s are also needed to determine whether the reduction was less than about 5% as the authors claimed. 

The statement in line 144-145 was based on two images shown in Fig 3b without quantification of these and other image data. Fig 3c and 5d need Y-axis.

Typos: Line 268, 'translation' should be 'translocation'. Line 283, 'by' should be 'but'. Line 296, should '3s' be '2s' since the label in the lower image in Fig 5b is 2s?

Comments on the Quality of English Language

As mentioned above.

Author Response

Reviewer 2

Q1 Overall, the presentation of the work is fine, most of the data are clearly shown and rightly interpreted, and the conclusions are mostly evidence based or reasonably argued. The working model presented at the end is interesting and communicates the take home message.

A1 Thank-you for your careful reading of this paper, and for your excellent summary.

Q2 (a)The Kd's in line 77-80 do not match the data in Fig 1b. (b) What is the [Ca2+] for the images shown in Fig 1a that also lacks a scale bar? (c) There is no Y-axis in Fig 1b. Fig 1c also lacks a scale bar.

A2  (a) The labels on the figure were wrongly placed but have now been corrected. (b) The Ca2+ in fig1a has now been added to the figure legend (1.3mM).  (c) the y-axis has been added.

Q3 Fig 2c. The fluorescence intensity in Y-axis ..  without the ticks   etc

A3 Fig 2c has been amended to show ticks, as requested

Q4 (a) The statement in line 144-145 was based on two images shown in Fig 3b without quantification of these and other image data. Fig 3c and 5d need Y-axis.

A4 (a) The quantified data from fig 3b has now been given in the legend. (b) The Y axis has been added to Fig 3c and 5d

Q5 Typos: Line 268, 'translation' should be 'translocation'. Line 283, 'by' should be 'but'. Line 296, should '3s' be '2s' since the label in the lower image in Fig 5b is 2s?

 A5 thank-you for pointing out these typos. They have now all been corrected

Round 2

Reviewer 1 Report

Comments and Suggestions for Authors

The authors have modified the text to address some of the points raised but reply only to selected quotes from the initial comments, ignoring some factual arguments and specific requests. I find this partial rebuttal unsatisfying and would like candid answers to the original points recopied below that were not properly acknowledged. 

Q1. Not acknowledged. “I understand that primary neutrophils are difficult to transfect, but why were macrophages chosen for the experiments in Fig. 5 and not the HL-60 cell line that can be differentiated into the neutrophilic lineage?” Please comment.  

Q1b. Not acknowledged. “At the minimum, the authors should repeat the Fluo4-zymosan experiments in RAW macrophages to show that they behave as neutrophils.” Please explain why the comment was not acknowledged and the experiment not done.

Q2. Not addressed. “Additional recordings are needed to properly document (…) the presence of sites of high Ca2+ concentrations around phagosomes (Fig. 5)”. Please explain why the comment was not acknowledged and the experiment not done. As such, the evidence for sites of high calcium concentrations around internalized phagosome is still based on n=1.

Q3. Dismissed as problematic. “The authors should provide functional evidence that calcium channels contribute to the drop in phagosomal Ca2+, for instance with inhibitors of TRPM2 (JNJ-28583113) or store-operated channels (GSK-7975A, SKF-96365 or lanthanum chloride).” Please cite relevant literature documenting that these compounds prevent phagocytosis. TRPM2 knock-out macrophages are still able to phagocytose (Di et al J Cell Sci 2017, cited Ref. 27). TRPM2 inhibition would thus not be expected to inhibit phagocytosis in neutrophils.

Q4. Not acknowledged. “Many functional studies report a functional ER in neutrophils (studies on ER stress effect in neutrophils, studies on store-operated calcium entry and STIM1/STIM2 function in neutrophils... etc.” Please comment. ER stress in neutrophils has been well documented (Vig et al 2022, https://doi.org/10.3389/fendo.2022.991632; Sule et al 2021, 10.1172/JCI137866 ; Hu et al 2015, 10.4049/jimmunol.1500073 ; Schirrmann et al 2022 10.1021/acsnano.2c03586 ; Lohdi et al 2015 10.1016/j.cmet.2014.12.002. ; reviewed briefly in Di Conza and Ho 2020 10.3390/cells903069). It seems unlikely that cells lacking ER should respond to ER stress induction in a similar manner as other cells, and augment expression or induce activation of ER stress markers such as BiP, PDI, CHOP, IRE1a. The revised statement that “neutrophils have no ER near the cell periphery” is difficult to reconcile with the genetic and functional evidence that STIM and ORAI proteins interact at ER-PM contact sites in neutrophils.

A4. I appreciate the value of images posted on Google but remain unconvinced. Lack of proof is not proof of absence, while ultrastructural evidence of ER in neutrophils is available (See picture attached). New imaging modalities such as Cryo-ET now provide fresh opportunities to debunk this old dogma.

Other comments:

page 7 lines 272-273 : “ TRPM2 Ca2+channels are found on neutrophil phagosomes [27] and carry currents of about 1pA [28] or about 107 ions/sec” The cited reference 27,  Di et al J Cell Sci 2017 shows the presence of TRPM2 on human and mouse macrophage phagosomes but not neutrophils, the study does not examine neutrophils at all. Reference 28, Starkus et al, J Physiol 2010 performs patch-clamp in whole-cell configuration in human neutrophils but the neutrophils are not phagocytosing. 

Comments on the Quality of English Language

Typos were fixed

Round 3

Reviewer 1 Report

Comments and Suggestions for Authors

The authors have adequately replied to the second set of comments and I would like to thank them for the fruitful discussion

I have no further suggestions for changes